# CNS Germ Cell Tumors: Molecular Advances, Significance in Risk Stratification and Future Directions

**DOI:** 10.3390/brainsci14050445

**Published:** 2024-04-29

**Authors:** Jiajun Zhou, Chenxing Wu, Shouwei Li

**Affiliations:** Department of Neuro-Oncology (No.6 Neurosurgery Department), Sanbo Brain Hospital, Capital Medical University, No.50. Yi-Ke-Song, Xiangshan, Haidian District, Beijing 100093, China; chow816@mail.ccmu.edu.cn (J.Z.); wucx@mail.ccmu.edu.cn (C.W.)

**Keywords:** germ cell tumors, chromosomal instability, gene mutation, DNA methylation, miRNAs, immune microenvironment

## Abstract

Central Nervous System Germ Cell Tumors (CNS GCTs) represent a subtype of intracranial malignant tumors characterized by highly heterogeneous histology. Current diagnostic methods in clinical practice have notable limitations, and treatment strategies struggle to achieve personalized therapy based on patient risk stratification. Advances in molecular genetics, biology, epigenetics, and understanding of the tumor microenvironment suggest the diagnostic potential of associated molecular alterations, aiding risk subgroup identification at diagnosis. Furthermore, they suggest the existence of novel therapeutic approaches targeting chromosomal alterations, mutated genes and altered signaling pathways, methylation changes, microRNAs, and immune checkpoints. Moving forward, further research is imperative to explore the pathogenesis of CNS GCTs and unravel the intricate interactions among various molecular alterations. Additionally, these findings require validation in clinical cohorts to assess their role in the diagnosis, risk stratification, and treatment of patients.

## 1. Introduction

### 1.1. Overview of Disease

Central Nervous System Germ Cell Tumors (CNS GCTs) are rare and histologically heterogeneous intracranial malignancies primarily affecting children, adolescents, and young adults [1,2,3]. According to the 2021 edition of the World Health Organization (WHO) classification of central nervous system tumors [4], CNS GCTs are classified into several subtypes, including germinoma, mature teratoma (MT), immature teratoma (IMT), teratoma with somatic-type malignancy, embryonal carcinoma (EC), choriocarcinoma (CC), yolk sac tumor (YST), and mixed germ cell tumors. Typical MRI images and histological images of central nervous system germ cell tumors are shown in Figure 1 and Figure 2, respectively. In clinical practice, the latter seven types are collectively referred to as non-germinomatous germ cell tumors (NGGCTs), which are distinguished from germinoma. Germinomas account for 64–75% of CNS GCTs, while more than half of NGGCTs are mixed germ cell tumors [5].

### 1.2. Clinical Management Strategies and Challenges

Currently, the classification of CNS GCTs still relies on traditional histomorphology, internationally recognized diagnostic criteria for CNS GCTs are lacking. The significance of neuro-imaging and clinical manifestations in the diagnosis and differential diagnosis of CNS GCTs is limited. Diagnosis primarily relies on tumor markers (human chorionic gonadotropin, HCG, and alpha-fetoprotein, AFP) in the blood and/or cerebrospinal fluid (CSF). A consensus on the diagnosis and treatment of CNS GCTs published in 2015 [6] stated: For suspected CNS GCTs patients, if β-HCG and AFP levels in the blood and/or CSF are within normal ranges, regardless of radiological findings, a biopsy is recommended to confirm the histopathology. If tumor markers are elevated and accompanied by typical radiological findings, histopathological examination may not be required, and treatment can be guided based on tumor marker levels. However, different countries and regions have different thresholds for defining “secreting” tumor markers. The American Children’s Oncology Group CNS Nongerminomatous Germ Cell Tumor Phase II Trial (ACNS1123) [7] sets HCG at 100 IU/L and AFP at 10 ng/mL, while the European SIOP GCT II Trial [8] sets HCG at 50 IU/L and AFP at 25 ng/mL. Furthermore, relying solely on tumor markers for diagnosis has significant limitations. There are histopathological types, such as embryonal carcinoma, that do not secrete tumor markers, as well as immature teratomas that can cause elevated AFP levels and germinomas that can cause mild elevation of HCG. Hence, the existing diagnostic methods are presently limited and pose challenges in achieving a definitive diagnosis. At present, the treatment of CNS GCTs involves a multimodal approach comprising surgery, chemotherapy, and radiotherapy. Germ cell tumors are generally sensitive to radiotherapy and chemotherapy, except for mature teratomas. Chemotherapy is often administered before radiotherapy to reduce the required dose and radiation field. Chemotherapy alone can provide relief for some cases, but it typically does not lead to long-term cure, especially for NGGCTs [9,10,11]. Surgery is primarily performed to obtain tissue for pathological diagnosis, particularly when tumor markers are negative, and to alleviate symptoms such as acute hydrocephalus or acute visual loss. Mature teratomas can often be effectively treated through surgical resection. In current clinical management, treatment grouping primarily relies on histopathological classification and prognosis. In Western countries, CNS GCTs (excluding mature teratomas) are typically classified into germinomas and NGGCTs for treatment purposes. In Japan, treatment strategies for these tumors are predominantly based on the three-tiered classification proposed by Matsutani et al. [12], which categorizes them as having a good, intermediate, or poor prognosis. 

Germinoma presents a favorable prognosis due to its high sensitivity to radiotherapy and chemotherapy, boasting a 10-year overall survival (OS) rate exceeding 90% [13,14,15,16]. In recent years, advancements in treatment regimens have led to gradual improvements in the prognosis of NGGCTs, achieving a 5-year survival rate of 70% or higher [3,13,14,15,16,17]. However, resistance to combination treatment regimens and disease recurrence still pose challenges for some patients [9,18,19], leading to a significantly poorer prognosis in recurrent cases [20,21,22]. The treatment approach for recurrent CNS GCTs lacks standardization and salvage treatment strategies post-recurrence have not substantially improved patient survival rates [23]. Further accumulation of experience and knowledge is needed to identify this subgroup of patients and develop targeted, novel treatment methods.

### 1.3. Past Endeavors, Existing Limitations, and Future Aspirations

In recent years, numerous prospective cohort studies have been conducted globally to optimize treatment outcomes while minimizing treatment-related burdens and long-term adverse effects for patients. These efforts aim to improve prognosis and enhance overall quality of life. However, current clinical management strategies, which are based on risk stratification, have limitations and face challenges in achieving precise and personalized diagnosis and treatment. Moreover, there is still a lack of established risk-stratified treatment guidelines to guide prospective clinical trials.

Therefore, our focus in managing CNS GCTs should extend beyond mere patient survival to enhancing prognosis and overall quality of life. Given that the majority of CNS GCT patients are children, adolescents, and young adults, the potential impact of chemotherapy and radiotherapy on neurological function, cognition, and the risk of secondary tumors must be carefully considered [24]. As molecular research on CNS GCTs, encompassing genetics and biology, continues to progress, these crucial findings offer new evidence and insights into disease mechanisms, aiding in diagnosis and prognostic prediction. However, despite these promising discoveries, their impact on current clinical management remains limited, as they are still in the laboratory or preclinical stages. By leveraging the genetic and biological molecular changes of the tumor in conjunction with clinical features, diagnosis can be further stratified into various subtypes across different histopathological types. This, in turn, facilitates related risk stratification and guides subsequent clinical management, offering significant benefits for patients, and exploring this avenue should be prioritized as an important research direction for the future.

This article aims to explore the guiding significance of these key findings in risk stratification and personalized diagnosis and treatment, as well as their future application value in clinical practice, by reviewing the recent advances in molecular research on CNS GCTs (Figure 3a). It offers new insights and approaches for the diagnosis of CNS GCTs and serves as a basis and support for subsequent clinical management.

## 2. Molecular Genetics

Genomic instability, recognized as a prominent characteristic and hallmark feature enabling human cancers, has been extensively studied [25]. It results in multilayered genetic alterations, including nucleotide changes at the gene level, structural alterations at the subchromosomal level, and gains and losses of entire chromosomes [26]. Numerous studies have investigated the genomic instability present in CNS GCTs, aiming to understand the mechanisms driving disease onset and progression.

### 2.1. Chromosomal Instability

CNS GCTs exhibit pronounced chromosomal instability, characterized by copy number gains, deletions, or structural alterations. Several relevant studies have been reviewed, revealing common chromosomal changes, including gains at 1q, 2p, 7q, 8q, 12p, 14q, 21q, and X, as well as losses at 1p, 4q, 5q, 9q, 10q, 11q, 13q, 17p, and 18p/q (Table 1) [13,27,28,29,30,31,32]. Patients’ clinical characteristics were associated with chromosomal instability features. Hirokazu Takami et al. pointed out that age is directly correlated with chromosomal instability, with significant trends of gains in 1q, 2p/q, 3q, 6q, 7p/q, 8p/q, 14q, 20p, 21q, and Y p/q, and losses in 9q, 15q, and 18q with increasing age [13]. Additionally, Dominik T. Schneider et al. also demonstrated that different age groups of CNS GCTs exhibit distinct chromosomal instability characteristics [29]. Furthermore, for CNS GCTs in different locations, research suggested that patients with lesions in the basal ganglia region tended to have more chromosome arm deletions, with an average loss of 12.3 chromosome arms, while patients with lesions in other locations had an average loss of 4.7 chromosome arms [13]. The relationship between chromosomal instability and age or gender was not observed in other research cohorts, awaiting confirmation from larger cohort studies. Whether the relationship between the characteristic of chromosomal instability and the site of onset implies different underlying pathogenic mechanisms requires further investigation.

In addition, different histological types of CNS GCTs exhibit distinct chromosomal instability features. Research indicated that chromosomal imbalances were more common in NGGCTs (average of 8.1 per tumor) compared to germinomas (average of 4.1 per tumor). Specifically, chromosomal gains were more prevalent in NGGCTs compared to germinomas (5.9 vs. 3.0 per tumor), while the frequency of chromosomal losses was similar (2.3 vs. 1.1 per tumor). Moreover, specific chromosomal imbalances were not associated with different malignant histological types [29]. C. H. Rickert et al. found that the most common chromosomal alterations differ among germinomas, mixed GCTs, teratomas, and yolk sac tumors [33]. Shintaro Fukushima et al. found that in germinomas, the most common chromosomal alterations were 1q, 21q, X gain, and 13q loss, while in NGGCTs, 12p, 21q gain, and X loss were more frequent [30]. Kaishi Satomi et al.’s study revealed a significant correlation between 12p gain and histological types: 12p gain was present in 12% of germinomas and 49% of NGGCTs. In NGGCTs, cases with malignant components (IT, YST, CC, and EC) showed a higher frequency of 12p gain compared to those without malignant components (63% vs. 17%). Furthermore, in mixed germ cell tumors, all pathological components shared the 12p gain status [34], indicating that 12p gain may play a crucial role as an early event in the development of the disease. The significant differences in chromosomal instability characteristics between germinomas and NGGCTs suggest potential mechanistic differences in their pathogenesis, with abnormal meiosis possibly playing a significant role in pure germinomas [31]. Compared to the high occurrence rate of 12p gain in testicular GCTs (77–88%), the incidence of 12p gain in CNS GCTs is significantly lower (20–57%) [34], suggesting that its role in the pathogenesis within the central nervous system may not be as crucial as in other sites of GCTs such as testicular GCTs. 

Studies have indicated that 21q gain and X gain were the most significant chromosomal alterations in CNS GCTs [30]. Bhattacharjee MB et al. also found that 21q gain was a unique genetic feature distinguishing pediatric CNS GCTs from other intracranial tumors in children [35]. Some congenital acquired disorders, such as Klinefelter syndrome (46, XXY) and Down syndrome (46, +21), have also been reported to be associated with CNS GCTs [36,37,38,39,40,41]. The higher incidence rate of CNS GCTs in these syndromes suggests that the chromosome X and chromosome 21 instability or the genes they carry may have a potential role in disease occurrence, but this has not been confirmed yet. If a specific chromosomal instability pattern could be identified, where key genes on that chromosome exert pathogenic effects and drive disease occurrence and progression, establishing a precise relationship between this subgroup of tumors and unique biological characteristics would be highly significant. This could facilitate the identification of patients within this subgroup and enable targeted treatments. Li B et al.’s latest study shed new light on this issue. The research observed amplifications of chromosome 12p12.1 (15.6%, 15/96), 4q12 (10.4%, 10/96), 22q11.21 (7.3%, 7/96), 1p13.2 (2.1%, 2/96), and 12q15 (2.1%, 2/96), as well as deletions of 11q24.2 (9.4%, 9/96) containing ARHGEF12 and BCL9L. Furthermore, compared to wild-type cases, the expression levels of the aforementioned target genes significantly increased in cases with copy number alterations [32], highlighting the functional impact of subchromosomal alterations. Additionally, sex chromosome aneuploidy was implicated in tumorigenesis and progression [36], with X chromosome polyploidy and hypomethylation considered mechanisms of malignant transformation [28]. These factors may partly explain the pronounced male predominance in CNS GCTs, but further research is needed to confirm gender-specific pathogenic mechanisms resulting from sex chromosome alterations. Chromosomal instability is generally associated with poor prognosis in patients, and this holds true in CNS GCTs as well. The presence of 12p gain could predict the presence of malignant components in NGGCTs, and its occurrence indicated a poor prognosis for patients [34]. Chromosomal aberrations such as increases in 2q and 8q and deletions in 5q, 9p/q, 13q, and 15q were also associated with poorer prognosis [13].

### 2.2. Gene Mutations and Signaling Pathway Alterations

While CNS GCTs are rare, recent research advancements have uncovered genetic mutations as significant driving factors for this disease. The occurrence of CNS GCTs was closely linked to genetic alterations within pathways such as the KIT/RAS pathway (MAPK pathway) and the AKT/mTOR pathway (PI3K pathway) [30,32,42,43]; the summary of gene mutations and the relevant pathways is presented in Table 2 and Figure 3b. Alterations in the KIT/RAS pathway represented the most common signaling pathway changes observed in CNS GCTs. Mutated genes within this pathway included KIT (21.5–33.3%), NRAS/KRAS (10.8–20%), CBL (5.2–11%), and NF1 (3%) [30,31,32,43]. KIT is an oncogene that encodes a transmembrane tyrosine kinase receptor. Functional mutations in the KIT gene can result in sustained activation of its transmembrane protein even in the absence of binding with the ligand stem cell factor (SCF), thereby activating downstream molecular signaling pathways such as the MAPK pathway or PI3K pathway. This leads to increased proliferation, migration, and resistance to apoptosis of tumor cells. Mutations in the RAS gene represented the second most common alterations, with mutations in KRAS and NRAS being the most frequent in CNS GCTs. The protein encoded by the RAS gene belongs to the GTPase family. Mutations in the RAS gene result in sustained activation of the RAS protein’s GTP-binding capacity, leading to activation of downstream signaling pathways and promoting cell proliferation and growth.

Compared to NGGCTs, germinomas exhibited differences in mutation characteristics within the KIT/RAS pathway. Research indicated that KIT mutations were present in 40% of germinomas and 6% of NGGCTs, while RAS mutations were present in 20% of germinomas and 3% of NGGCTs. Furthermore, in the cohort, among the three cases of NGGCTs with KIT or RAS mutations, two were mixed GCTs with combined germinoma components [30]. Therefore, alterations in the KIT/RAS pathway may play a more significant role in the pathogenesis of germinomas, and non-mixed subtype NGGCTs may have a lower dependence on KIT/RAS signaling alterations. In another study, after dissecting and analyzing the mutation features of the various pathological components of mixed GCTs, it was found that the germinoma component and NGGCT component shared common KIT/RAS mutation characteristics but with different levels of methylation [42]. This suggests that the different components of mixed GCTs may originate from the same precursor cells, and the occurrence of KIT/RAS mutations precedes the appearance of different histological components, possibly representing the initial defining event in these precursor cells. Subsequent mechanisms, such as epigenetic regulation, may then drive their growth and differentiation into different pathological types. Specifically, KIT mutations and KRAS/NRAS mutations have been described as mutually exclusive genetic events in CNS GCTs [30,31], meaning they were rarely coexistent within the same tumor. This strongly suggests that these genes may exert their pathogenic effects through the same pathway, and alteration in either gene alone is sufficient to drive the occurrence and development of the disease. The protein encoded by the CBL gene has been shown to exert negative regulation on receptor tyrosine kinase proteins (RTKs), including KIT, by mediating ubiquitination and degradation [31]. Somatic mutations in CBL occurring in CNS GCTs result in loss of its negative regulation on KIT, leading to upregulation of signaling pathway function and contributing significantly to KIT overexpression. Additionally, NF-1 acts as a negative regulator of the MAPK pathway, and mutations in NF-1 have also been observed in CNS GCTs.

Another important genetic event occurring in CNS GCTs involves alterations in the PI3K pathway, which constitutes another downstream pathway of the KIT receptor, involving AKT and mTOR. Changes in this pathway represented 12.9–19% of CNS GCTs, with one or more alterations in components such as PIK3C2B, PIK3R2, AKT1, mTOR, and PTEN, among others [13,31,43]. mTOR, short for mammalian target of rapamycin, is an atypical serine/threonine protein kinase that interacts with two protein complexes, mTORC1 and mTORC2, to promote cell growth, proliferation, and survival. Mutations in mTOR have been observed in 6.5–8% of CNS GCTs [31,43]. AKT, also known as protein kinase B, is a serine/threonine protein kinase that acts downstream of PI3K to regulate cellular metabolism and promote cell growth, proliferation, and survival upon activation. Wang et al. observed an amplification rate of 19% for AKT1 in CNS GCTs, and in 75% of cases with increased AKT1 copy numbers, KIT, KRAS, and NRAS were wild-type, indicating the independent role of the AKT/mTOR pathway in the pathogenesis of CNS GCTs apart from the KIT/RAS pathway. Unlike the more frequent alterations observed in the KIT/RAS pathway in germinomas, the occurrence rate of mutations in the AKT/mTOR pathway was similar between germinomas and NGGCTs [43]. Additionally, mutations in PTEN were present in 2% of CNS GCTs [31]. The protein product encoded by the PTEN gene exerts negative regulation in the PI3K pathway, and mutations in PTEN are also one of the reasons for the activation of the PI3K pathway.

Overall, more than 49% of CNS GCTs harbored at least one somatic mutation in genes of the KIT/RAS pathway or the AKT/mTOR pathway [31,43], demonstrating the pivotal role of these genetic alterations in tumorigenesis and possibly representing the earliest occurring and pathogenic driving events. A considerable portion of CNS GCTs did not harbor genetic alterations in the KIT/RAS or AKT/mTOR pathways. However, regardless of the KIT mutation status, almost all germinomas showed positive expression in immunohistochemistry (IHC) for KIT, and approximately half of NGGCTs also exhibited KIT-positive expression in IHC [30]. This suggests the existence of other mechanisms leading to functional activation of KIT or alterations in other yet unidentified members of the same signaling pathway contributing to their development. The latest findings have illuminated the pathogenic mechanisms underlying signaling pathway alterations and provided partial insights into the reasons for the heightened activation of the MAPK pathway. In 15 cases with copy number gains at chromosome 12p12.1 and 7 cases at 22q11.21, Li B et al. [32] observed high-level amplification (CN > 10) in 6 and 5 cases, respectively. The overlapping regions on 12p12.1 and 22q11.21 in these samples contain known oncogenes KRAS and CRKL. CRKL has been demonstrated to activate the RAS signaling pathway [44,45]. Meanwhile, the study combined RNA-seq and FISH analyses to compare the transcriptional levels and the expression levels in tumor cells; these cases showed significantly increased transcription levels of KRAS and CRKL, as well as consistent high abundance signals in the nucleus. This not only highlights the potential driving role of the RAS pathway in the occurrence and development of CNS GCTs but also further suggests that KRAS amplification may occur in the form of extrachromosomal DNA (ecDNA), potentially representing a novel oncogenic mechanism in CNS GCTs. Furthermore, the study identified a potential new driver gene: USP28. The protein encoded by USP28 is a ubiquitin-specific protease belonging to the deubiquitinating enzyme (DUB) family. Mutations in USP28 were observed in 8 samples (8.3%) of CNS GCTs, with the majority being truncating mutations resulting in loss of USP28. Further analysis revealed that truncating mutations in USP28, compared to the wild type, led to increased expression of BRAF and phosphorylated ERK. This confirms that loss-of-function mutations in USP28 can induce MAPK pathway activation by stimulating the fibrosarcoma-ERK pathway. In vitro experiments also showed that 293T cells with mutations in USP28 were responsive to treatment with trametinib, a selective inhibitor of MEK1/2. These findings deepen our understanding of the MAPK pathway activation mechanism and suggest the presence of a patient subgroup with USP28 mutations, which could potentially be targeted with therapy aimed at the MAPK pathway.

Gene and signaling pathway alterations were also linked to clinical characteristics and patient outcomes. Research by Hirokazu Takami et al. revealed gender disparities in gene mutations, with MAPK pathway mutations detected in half of male cases but less frequently in females [13]. This suggests gender-specific disparities in the development and progression of germ cell tumors, potentially influenced by the Y chromosome in males and alternative pathogenic pathways in females, independent of MAPK pathway alterations. Furthermore, tumors located in the basal ganglia and ventricles appeared to frequently exhibit alterations in the PI3K/mTOR pathway compared to other locations [13], suggesting distinct mechanisms of disease occurrence between atypical (basal ganglia, ventricles, etc.) and typical (sellar region, pineal region) intracranial sites. Shintaro Fukushima et al. observed in their study that germinomas with KIT/RAS alterations demonstrated a trend towards shorter progression-free survival (PFS), although the results did not reach statistical significance [30]. Research by Koichi Ichimura et al. suggested that germinomas with MAPK pathway mutations showed a trend towards prolonged OS, albeit statistically nonsignificant, while the opposite trend was noted in NGGCTs. Cases with alterations in the PI3K/mTOR pathway exhibited a trend towards shorter OS and PFS in the germinoma subgroup and all CNS GCT cases combined, although this trend was not observed in the NGGCT subgroup [43]. The characteristics mentioned above may be affected by inherent limitations of retrospective studies, including variability in past treatments received by each patient and the limited number of cases in each group, particularly in the various histological subtypes of NGGCTs. These limitations can cast uncertainty on conclusions regarding the impact of genetic factors on the clinical course of CNS GCTs. Future prospective studies with larger case numbers are warranted to investigate their effects and yield more accurate results.

Additionally, CNS GCTs have been associated with additional genetic mutations. Keita Tershima et al. demonstrated frequent alterations in the PRDM14 gene (13/27, 47%), as well as CCND2 (14/27, 51%) and RB1 (13/27, 47%) in CNS GCTs [27]. PRDM14 functions as a transcription factor involved in specific transcriptional regulation in primordial germ cells, while CCND2 and RB1 encode proteins that regulate the cell cycle, influencing cell cycle progression through the Cyclin/CDK-RB-E2F pathway and affecting cell proliferation. These gene alterations suggest potential roles for transcriptional regulation specific to primordial germ cells and the Cyclin/CDK-RB-E2F pathway in the pathogenesis of CNS GCTs [27]. Cell cycle pathways (5.2%) and epigenetic regulation pathways (3.1%) have also been implicated in the development of CNS GCTs [32]. Yaser Atlasi et al. reported an association of the Wnt/β-catenin pathway with embryonal carcinoma [46], but its alterations and effects in CNS GCTs have not been clarified yet, awaiting further research for elucidation. BCORL1, a transcriptional corepressor located on the X chromosome, was found to have functional loss mutations in 7.3–9.7% of CNS GCT cases [31,32]. Additionally, a chromatin-modifying gene, JMJD1C, was observed to have mutations in 16.1% of cases [31]. Interestingly, BCORL1 and JMJD1C may interact with androgen receptors during the puberty-related increase in testosterone levels [31]. The functional loss mutation of BCORL1 leads to the loss of its negative regulation on androgen receptors, combined with the unique changes of JMJD1C in CNS GCTs, suggesting that their mutations may contribute to the tumor’s specific male predilection and age-related characteristics of peak incidence during puberty. The specific functions and mechanisms of these genes await further confirmation through future research. JMJD1C gene alterations were also observed to be significantly enriched in the Japanese population [31], suggesting a potential impact of genetic alterations in CNS GCT on racial and regional differences in its epidemiology.

Alterations in the KIT/RAS pathway and the AKT/mTOR pathway also provide a basis for targeted therapy. Targeted tyrosine kinase inhibitors (TKIs) against KIT, Erk1/2 inhibitors, AKT kinase inhibitors, dasatinib targeting CBL, and the mTOR-targeting agent pp241 (Torkinib) are anticipated to be utilized clinically in the future. Some efforts have been undertaken toward targeted therapy in CNS GCTs. Abu Arja et al. reported the utilization of brentuximab–vedotin (an antibody-drug conjugate targeting CD30) in the treatment of a patient with Down syndrome and intracranial embryonal carcinoma [47]; Osorio DS et al. shared their experience treating 6 cases of CNS GCTs using the second-generation TKI, dasatinib [48]. Schultz, KA et al. reported a successful case of treating a pediatric intracranial growing teratoma syndrome using the selective reversible CDK 4/6 inhibitor, PD0332991 [49]. However, a study by the Abramson Cancer Center at the University of Pennsylvania initiated a clinical trial in 2009 (NCT01037790) to investigate the efficacy and side effects of PD-0332991 in refractory solid tumors [50]; the results were relatively disappointing in the treatment of cisplatin-refractory and unresectable adult CNS GCTs. Among the 26 evaluable patients, 17 exhibited stable disease (SD), while 9 showed progressive disease (PD). The above experiences suggest the clinical potential of targeted therapy, and in the future, with larger cohorts and the simultaneous application of more precise strategies based on corresponding molecular alterations to distinguish respective subgroups of patients, there is optimism for the development of novel and more precise targeted treatment strategies.

For germinomas, precision treatments targeting molecular genetic alterations have the potential to minimize radiation therapy dosage and area while also influencing chemotherapy’s role. This approach aims to mitigate neurocognitive and neurological side effects, thereby enhancing the prognosis and quality of life in young patients. In cases of NGGCTs with poor prognoses, a comprehensive treatment strategy combining targeted therapy tailored to genetic alterations with conventional radiation, chemotherapy, and surgery may yield the most favorable outcomes. Moreover, targeted therapy holds promise for patients facing treatment resistance and recurrent disease, often associated with dismal prognoses.

In summary, recent studies have unveiled potential pathogenic mechanisms underlying CNS GCTs through genetic alterations. For this diverse and intricate tumor group, investigating chromosomal imbalances, gene mutations, and aberrant signaling pathways is paramount. The integration of these alterations with their clinical implications and their accurate reflection of tumor biology, which necessitates further elucidation in future studies, holds the potential to refine the classification of CNS GCTs based on genetic characteristics. Moreover, incorporating genetic profiles into prognosis assessment and validating their role in treatment within prospective clinical cohorts can lead to more nuanced and personalized clinical management strategies, offering significant benefits for patients.

## 3. Gene Expression and Transcriptomes

Recent analyses of expression and transcriptome in CNS GCTs have unveiled the biological characteristics of various histological types of these tumors and highlighted potential key biological processes and critical genes implicated in their pathogenesis, laying a solid foundation for further investigation into these tumors.

Several transcriptome studies on CNS GCTs have revealed distinct expression patterns: germinomas exhibited high expression of early primordial germ cells (PGCs) markers and features of mitosis/meiosis, whereas NGGCTs showed elevated expression of genes associated with tissue/organ development, the Wnt/β-catenin signaling pathway, and epithelial-mesenchymal transition [51,52]. Integration of expression data from GCTs and normal embryonic cell development suggested that germinomas belong to the PGC lineage, while NGGCTs shared similarities with embryonic stem cells (ESCs) [51]. The research findings strongly suggest the potential origin of CNS GCTs, with germinomas closely associated with the early stages of germ cell development, specifically PGCs, while NGGCTs are linked to cells in a more differentiated state, progressing towards tissue and organ development. IHC protein expression results also validated these characteristics of CNS GCTs: pluripotency markers such as KIT, OCT3/4, NANOG, TFAP2c, KLF4, and DPP4 were highly expressed in germinomas, along with germ cell-specific genes such as MAGEA4, NY-ESO-1, and TSPY [52,53], underscoring the intimate relationship between GCTs and primordial germ cells.

In NGGCTs, the high expression of two key regulatory factors involved in epithelial-mesenchymal transition (EMT), SNAI2 (SLUG) and TWIST2, suggests their potential role in promoting mesenchymal transition, thereby enhancing the invasive and migratory capabilities of cells. This could contribute to tumor malignancy and stemness [52]. Given the concurrent overexpression of the Wnt/β-catenin signaling pathway in NGGCTs [46,52] and its involvement in EMT regulation, it is plausible that certain subgroups of NGGCTs exhibit these characteristics; this point has been further substantiated in the latest study [32]. Li B et al. proposed a novel molecular classification for CNS GCTs based on gene set enrichment analysis of RNA-seq data from 92 samples in the cohort. Subtype 1 was characterized as immune-hot, with a strong activation of immune-related pathways, primarily composed of germinomas. Tumors in this group predominantly originated from the hypothalamic–pituitary region and/or pineal region, and patients tended to have a higher age of onset compared to the other two groups. Subtype 2, characterized as MYC/E2F type, exhibited enrichment in cell cycle-related pathways and high expression of MYC/E2F target genes, primarily composed of germinomas. Patients in this subtype tended to have a younger age of onset. Subtype 3, identified as SHH type, demonstrated high activity in genes related to the SHH pathway and EMT signaling pathway. This group included a minority of germinomas (21.3% of all germinoma cases) and the majority of NGGCTs (70.6% of all NGGCTs cases), with a predominance of male patients. This marks the first instance of proposing a molecular classification of CNS GCTs. Such an endeavor holds profound significance in enhancing our comprehension of the disease and broadening the scope of clinical treatment strategies.

Studies by Ryo Nishikawa et al. indicated that p52 protein was expressed in 94% of CNS GCTs, while p21WAF1/Cip1 protein was expressed in 20% of cases. Notably, p21WAF1/Cip1 expression was absent in germinomas but present in 27% of teratomas and 80% of malignant NGGCTs. The overexpression of p21WAF1/Cip1 protein may correlate with reduced sensitivity to radiotherapy and chemotherapy, indicating a poorer prognosis [54]. However, the mechanisms and specific biological significance of the high expression of p52 and p21WAF1/Cip1 require further elucidation. In addition, studies by James E. Korkola et al. and Hsei-Wei Wang et al. indicated that gene expression characteristics could predict the prognosis of CNS GCTs. The results suggest that GCTs with a favorable prognosis express genes related to immune response and other immune functions, as well as genes involved in inhibiting differentiation and proliferation. In contrast, GCTs with a poorer prognosis exhibited expression of genes and pathways associated with active development and differentiation, particularly those related to neurodevelopment [52,54].

In summary, the aforementioned findings underscore the close association between tumor development and the diverse biological characteristics underlying the pathogenic mechanisms of CNS GCTs, a group of tumors characterized by significant heterogeneity. Developmental processes, differentiation, and the tumor microenvironment exert significant influences on tumor biology and subsequent therapeutic responses like radiotherapy and chemotherapy. Further foundational research is necessary to establish the role of expression profiles in risk stratification, prognosis prediction, and treatment response monitoring in GCT, potentially shaping future clinical management strategies.

## 4. Epigenetics

In addition to chromosomal instability and gene mutations driving tumor formation and development, another distinct mechanism involves epigenetic regulation of gene expression, known as “non-mutational epigenetic reprogramming”. This process is also central to embryonic development, differentiation, and organogenesis [25]. Advancements in understanding epigenetic alterations in CNS GCTs have unveiled the significance of non-mutational epigenetic mechanisms in both tumor initiation and progression.

### 4.1. DNA Methylation

In CNS GCTs, distinct DNA methylation features were observed across different histopathological components. Shintaro Fukushima et al. revealed that over 59% of germinomas exhibited overall low methylation, with no prominent peaks in tumor-specific methylation probes, while 86% of NGGCTs displayed high methylation and showed abundant tumor-specific methylation probes. Interestingly, microdissection and methylation analysis of each component in mixed germ cell tumors demonstrated that while each component shared the same somatic mutations, there were significant differences in methylation features between germinoma and NGGCT components. Germinoma components clustered within low-methylation clusters, whereas NGGCT components clustered within high-methylation clusters [42]. The low methylation profile of germinomas bears a striking resemblance to that of PGCs, further supported by the comparable methylation levels between germinoma genomic imprints’ differentially methylated regions and those of PGCs [42,55]. This strongly suggests that germinomas may originate from PGCs, while NGGCTs may arise from more primitive stem cells or cells closely resembling embryonic stem cells, indicating fundamental differences in cell origin or differentiation between the two. For mixed GCTs, they were more likely to arise from the same group of cells with initial mutations in the MAPK/PI3K pathways, followed by the acquisition of NGGCT components from germinomas due to epigenetic alterations. This illustrates that changes in signaling pathways may typically precede epigenetic dysregulation as pathogenic factors in the process of CNS GCT development.

Germinomas exhibited highly hypomethylated genomes across the entire genome, a recognized cause of genomic instability, rendering these tumors prone to progression. DNA methylation typically involves epigenetic reprogramming during germinoma development; hence, aberrant methylation patterns may play a crucial role in the pathogenesis of germinomas. Studies have shown that in CNS GCTs, the methylation profile was significantly associated with changes in the MAPK/PI3K pathways and chromosomal instability, with overall hypomethylation being significantly correlated with severe chromosomal instability in germinomas [42]. Therefore, it is believed that MAPK and/or PI3K pathway alterations, overall low DNA methylation, and chromosomal instability are the three key factors contributing to the pathogenesis of germinomas [42].

LINE1 belongs to the human retrotransposon family, and its transpositional activity is regulated by DNA methylation. The methylation level of LINE1 can reflect the methylation status of the entire genome [56]. In a study, the LINE1 methylation level in germinomas (median 31.2%) was significantly lower than that in normal tissues (64.4%). In some cases of germinomas, the methylation level was below 20%, and in two cases, complete demethylation of LINE1 was observed [42]. JN Jeyapalan et al. also reported a significant decrease in LINE1 methylation levels in germinoma samples compared to normal tissues (32% vs. 67%) [57]. Such a phenomenon has not been observed in other malignant tumors or normal tissues. Therefore, in germinomas, the overall hypomethylation of the tumor may induce genomic instability through LINE1 hypomethylation, thereby promoting tumorigenesis and progression to some extent [42].

In addition, the unique methylation characteristics of CNS GCTs also influenced gene expression. A characteristic of germinomas was the high expression of genes associated with PGCs and meiosis. Studies have shown that compared to NGGCTs, these genes exhibited lower methylation levels and higher expression levels in germinomas [51], indicating the promoting effect of unique DNA methylation characteristics on the disease. There was also evidence suggesting that methylation characteristics in germinomas may provide certain prognostic indications. Germinomas with overall hypomethylation tended to have a shorter PFS compared to those with partial hypomethylation or hypermethylation, although the difference did not reach statistical significance. No differences were observed in OS among different methylation level groups [42]. Further validation is needed to assess the impact of different methylation levels on patient prognosis.

Therefore, current evidence suggests that CNS GCTs exhibit distinct methylation characteristics across different histological components, playing a significant role in the onset and progression of the disease. Future efforts should aim to elucidate the specific mechanisms by which DNA methylation promotes tumor development in different histological types of tumors and explore personalized risk stratification based on these features in conjunction with clinical information. This approach could facilitate the development of novel targeted therapies to optimize the clinical management strategies for CNS GCTs.

### 4.2. Non-Coding RNAs

Non-coding RNAs play crucial roles in regulating various key processes involved in tumor initiation and progression. In recent years, research on non-coding RNAs in GCTs has primarily focused on microRNAs (miRNAs). miRNAs are short, single-stranded, non-coding RNAs that regulate gene expression by translational repression and/or mRNA degradation. They play important roles in cellular processes such as development, differentiation, apoptosis, tumor formation, and metastasis. Additionally, some researchers believe that miRNAs serve as both oncogenes and tumor suppressor genes, playing critical roles in tumor development [58].

Multiple studies have reported differences in the miRNA profiles between germinomas and NGGCTs. Research by Hsei-Wei Wang et al. indicated that most differentially expressed miRNAs were downregulated in pediatric intracranial germinomas, while the miR-142-5p and miR-146a were upregulated in pediatric intracranial NGGCTs [52]. Tsung-Han Hsieh et al.’s study, utilizing smRNA-seq detection, identified 27 new miRNAs with differential expression between pediatric intracranial germinomas and NGGCTs [59]. Research by Roger D. Palmer et al. on gonadal GCTs demonstrated the upregulation of miR-371-373 and the miR-302 cluster in malignant GCTs [58]. Matthew J. Murray et al.’s study indicated that the miR-302 cluster was overexpressed in all malignant GCTs and showed further overexpression in YST compared to germinomas [60]. These findings also suggest differences in the drivers between germinomas and NGGCTs.

Moreover, in GCTs, the expression of miRNAs profoundly influences the biological characteristics of tumors. Research suggested that in testicular germ cell tumors, miR-372 and miR-373 may impact the functionality of the p53 pathway in tumorigenesis by directly suppressing the expression of the tumor suppressor LATS2 [61]. Another study revealed that miRNA-371-373 and the miR-302 cluster were commonly overexpressed in malignant GCTs and exert their functions by downregulating downstream mRNA involved in important biological pathways [58]. Furthermore, in CNS GCTs, research has shown that overexpression of miR-214-3p promoted resistance to cisplatin by targeting the pro-apoptotic protein BCL2L11 [59]. Additionally, studies have unveiled the relationship between miRNAs and the corresponding levels of gene expression regulation in CNS GCTs. In germinomas, the expression levels of RUNX1T1 and THRB were negatively correlated with miR-145a expression, while the levels of NRP1, SVIL, and PDGFRA were negatively correlated with miR-142-5p expression. RUNX1T1 may be a target of miR-142-5p and miR-146a. In NGGCTs, miR-218 was negatively correlated with the expression of downstream targets [52]. A study has also suggested that differential miRNA expression may contribute to the relatively invasive behavior of YSTs [39]. When considering the role of miRNAs in intracranial and extracranial GCTs, a complex miRNA–mRNA genetic network has been identified [29]. Further research is needed to understand the precise biological roles of different miRNAs in various histological types and their clinical significance, aiding in the identification of new therapeutic targets characterized by miRNA alterations.

miRNAs exhibited high stability in extracellular fluids [62,63] and could be detected in serum and cerebrospinal fluid using qPCR, suggesting their potential as novel biomarkers. For instance, in the case of intracranial malignant GCT, elevation of miR-371a-3p levels in the cerebrospinal fluid preceded histological confirmation by 2 years, indicating the potential for early disease identification and diagnosis using miRNAs [64]. Research by Klaus-Peter Dieckmann et al. demonstrated the diagnostic value of miR-371a-3p, with a sensitivity of 88.7% and a specificity of 93.4% [65]. Matthew J. Murray et al. developed an miRNA panel comprising four miRNAs (miR-371a-3p, miR-372-3p, miR-373-3p, and miR-366-3p), demonstrating its high sensitivity and specificity for diagnostic purposes [66]. Additionally, miRNA levels play a crucial role in distinguishing intracranial malignant GCT from conditions like Langerhans cell histiocytosis [64]. 

In the future, non-invasive diagnostic methods targeting specific miRNA alterations show promise for early detection and diagnosis of CNS GCTs. However, specific miRNAs corresponding to histological types or risk stratification require further discovery and validation. Beyond diagnosis, miRNAs also play a role in monitoring treatment response. Several studies have demonstrated their potential value in monitoring residual lesions post-treatment [65,67,68]. Particularly, miR-371-3p showed promise in detecting GCT recurrence early, surpassing traditional markers like AFP and HCG in sensitivity [64,65,68]. Establishing standardized measurement methods and thresholds would be crucial to fully harness the potential applications of miRNAs in diagnosis, treatment response assessment, and disease recurrence evaluation. Integrating miRNA testing into disease follow-up protocols may enhance clinical management strategies.

In summary, elucidating the link between tumor methylation profiles and clinical outcomes, treatment responses, and prognosis may facilitate the categorization of diverse risk subgroups based on varying methylation patterns, allowing for tailored treatment strategies. While the high sensitivity and specificity of miRNAs position them as promising novel molecular markers, epigenetic characteristics are anticipated to hold significant implications for future clinical interventions.

## 5. Immune Infiltration in Tumor Microenvironment

Numerous factors contribute to the pathophysiology of CNS GCTs, including molecular genetic changes, epigenetic alterations, and the tumor microenvironment. Histologically, germinomas often exhibited abundant lymphocytic infiltration, termed the “dual-cell pattern”. This microenvironment primarily comprised CD3+ T cells, CD4+ helper T cells, CD8+ cytotoxic T cells, and B cells, including plasma cells, with fewer NK cells, monocytes, and other cell types present [69,70]. NGGCTs exhibited a more diverse immune cell composition compared to germinomas. Takami et al.’s research revealed variability in the tumor cell-to-immune cell ratio in germinomas, with cases showing lower tumor cell counts having abundant immune cell infiltration, while cases with higher tumor content had fewer immune cells. In contrast, NGGCTs displayed a richer infiltration of tumor cells, with activated NK cells, monocytes, and M2 macrophages being more prevalent [51]. These distinct characteristics in the tumor microenvironment highlight differences in the pathogenesis between germinomas and NGGCTs, offering insights for histological classification in diagnosis and the development of novel therapeutic strategies. The infiltration patterns of immune cells were linked to the clinical phenotype and patient characteristics. In germinomas, the extent of immune cell infiltration mirrored the proportion of tumor cells in the tissue. Some studies have proposed that tumor content influences the prognosis of germinomas [70,71]. Specifically, germinomas with greater immune cell infiltration tended to have a more favorable prognosis, suggesting a potential for grouping patients based on tumor and immune cell content. Cases with higher immune cell content and lower tumor cell content might benefit from less intensive treatment. However, further research and validation are necessary to confirm these findings.

The significant role of the tumor microenvironment in promoting immune suppression has been highlighted, leading to the progression or metastasis of various cancers [72]. Programmed cell death protein 1 (PD-1), a crucial immune checkpoint inhibitory receptor belonging to the CD28 family, is primarily expressed on T lymphocytes. It plays a vital role in immune tolerance and immune evasion of tumor cells. Its ligand, programmed cell death-ligand 1 (PD-L1), is expressed on tumor cells and various immune-related cells, including T cells, antigen-presenting cells, and macrophages. The interaction between PD-1 and PD-L1 induces T cell dysfunction, weakening the anti-tumor immune system as part of the immune checkpoint axis, thereby allowing tumor cells to evade the host’s anti-tumor immune response. Several studies have examined PD-1/PD-L1 expression in CNS GCTs, highlighting the pivotal role of the PD-1/PD-L1 axis in these tumors. Research by Bin Liu et al. and Hirokazu Takami et al. revealed that 93.8–96% of immune cells express PD-1, while 73.5–92% of tumor cells express PD-L1 in CNS GCTs [70,73]. Li B et al. also noted substantial upregulation of immune checkpoint genes, including PD-1, PD-L1/L2, cytotoxic T-lymphocyte-associated protein 4 (CTLA-4), lymphocyte activation gene 3 (LAG3), and indoleamine 2, 3-dioxygenase 1 (IDO1), in the subset of CNS GCTs exhibiting an evident immune-hot microenvironment [32]. Moreover, a close association between PD-1 expression and the density of CD3+, Foxp3+, and CD8+ lymphocytes, as well as the ratio of Foxp3+/CD4+ lymphocytes [73], in line with the findings of Pia Zapka et al., which demonstrated the dominance of CD3+ T cells in the tumor microenvironment, with an elevated regulatory T cell population and a significant presence of PD-1-positive immune cells [69], suggests an immune-suppressive state characterized by T cell dysfunction in CNS GCTs. The PD-1/PD-L1 axis likely contributes to the disruption of the anti-tumor immune response, leading to immune tolerance in these tumors. These findings suggest that the immune-suppressive nature of CNS GCTs might contribute to resistance against conventional therapies. Immune checkpoint inhibitors hold promise for potential application in CNS GCTs. A study reported a case wherein camrelizumab, a humanized selective IgG4-kappa monoclonal antibody against PD-1, was used as monotherapy for a patient experiencing a third recurrence of germinoma. Following one month of treatment (200 mg every 2 weeks), substantial regression of the patient’s spinal lesions was observed. This treatment response persisted for approximately six months until new metastatic lesions emerged [32]. A clinical trial investigating the use of immune checkpoint blockade drugs in the treatment of CNS GCTs has commenced [74], although definitive results are pending.

Furthermore, research has shown a correlation between the composition of immune cells in CSF and tumor type. Germinomas typically exhibited a significant presence of lymphocytes, whereas NGGCTs tended to have a notable abundance of monocytes. Optimal cutoff values for immune cell scores were calculated to distinguish between germinomas and NGGCTs, revealing that monocytes ≥ 21% or lymphocytes ≤ 52% are optimal thresholds for diagnosing NGGCTs (with a sensitivity of 90.9% and specificity of 77.8%) [75]. This highlights the potential diagnostic and classification utility of immune cell scores in CSF, suggesting promising prospects for their future application in disease diagnosis and risk stratification.

In summary, the diverse immune infiltration patterns observed across different cases hold promise as a means to predict treatment response and prognosis in patients. With the notable upregulation of immune checkpoint markers in CNS GCTs, immunotherapy targeting these checkpoints is anticipated to augment current treatment approaches, potentially benefiting patients with suboptimal responses to standard therapies or those facing recurrence.

## 6. Future Prospects and Conclusions

In recent years, advancements in genomic and epigenetic technologies have significantly enhanced our understanding of the pathogenesis of CNS GCTs; these include insights into chromosomal alterations, gene mutations, epigenomic modifications, transcriptomics, miRNA profiles, and the tumor immune microenvironment, leading to significant progress in our knowledge base. However, despite these advancements, translating these discoveries into clinically applicable diagnostic methods or treatment strategies remains elusive. The limited availability of tumor tissue samples and the absence of in vitro cell models pose considerable challenges and need to be addressed in the future. More efforts should focus on two main aspects. Firstly, there is a need for in-depth and comprehensive research into the detailed pathogenesis of CNS GCTs, exploring the specific mechanisms and interactions of genetic changes, epigenetic alterations, transcription, translation, and the tumor microenvironment. Secondly, it is essential to incorporate relevant findings into prospective clinical trial cohorts with a sufficient number of cases and to adopt new treatment methods. This will clarify whether different genetic or biological features can be used for patient risk stratification at the time of diagnosis, allowing for the early identification of patients belonging to molecularly characterized subgroups. However, CNS GCTs remains a relatively rare disease, especially in Western countries. Therefore, larger cohorts are more likely to be established in Asia, particularly in East Asia. It remains to be seen whether the combination of new treatment methods with conventional radiotherapy and chemotherapy regimens can improve patient prognosis and reduce the long-term adverse effects typically associated with previous treatments. Additionally, it is important to investigate whether patients with unique molecular features can benefit from lower-intensity treatment approaches. For patients with refractory or recurrent CNS GCT who are resistant to conventional treatment methods, the potential benefits of new therapies targeting their molecular characteristics also warrant further investigation.

## Figures and Tables

**Figure 1 brainsci-14-00445-f001:**
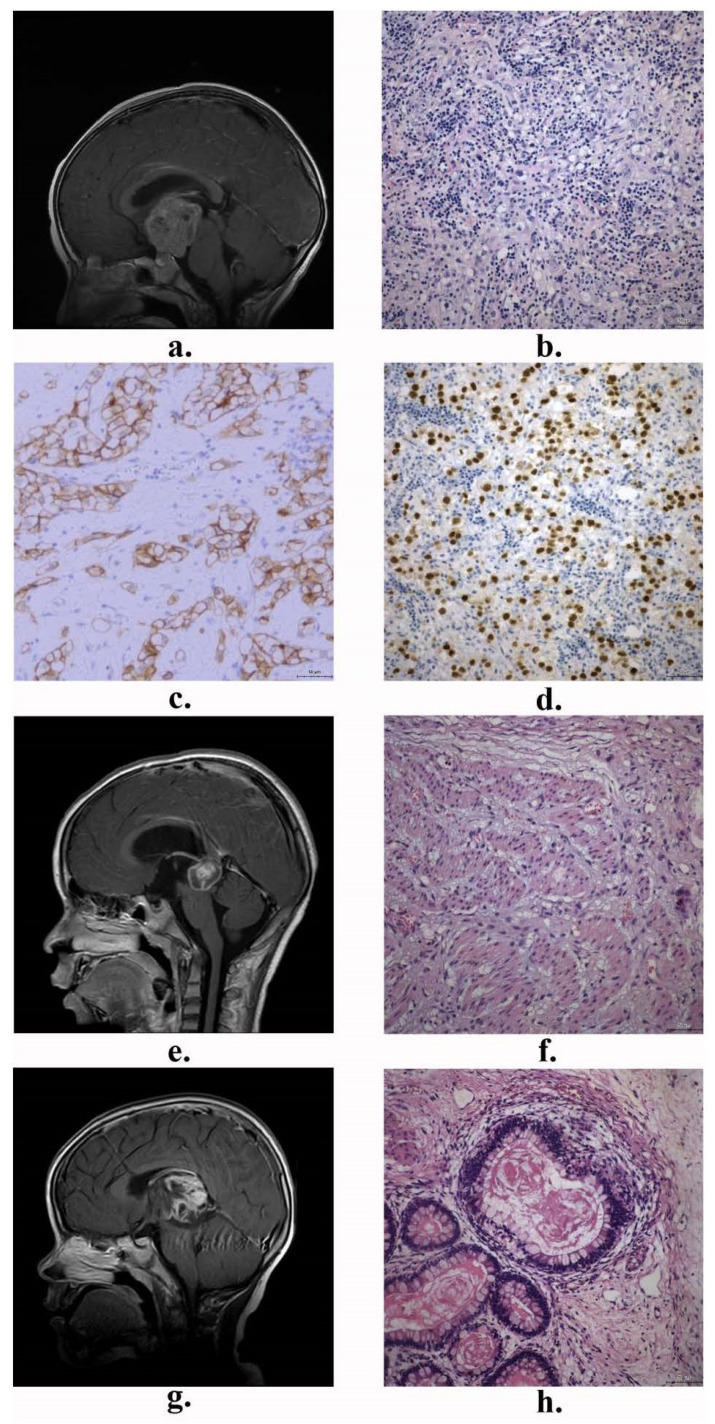
Typical magnetic resonance imaging (MRI) pictures and histopathological pictures of central nervous system germinomas and teratomas: (**a**) T1-weighted gadolinium-enhanced MRI from a patient with germinoma; (**b**) The lesion stained with hematoxylin and eosin (HE); (**c**,**d**) Immunohistochemistry (IHC) staining images of c-Kit (CD117) and OCT3/4 expression; (**e**,**g**) T1-weighted gadolinium-enhanced MRI from the patient with mature teratoma and immature teratoma; (**f**,**h**) HE staining images of mature teratoma and immature teratoma. (HE and IHC pictures: 200×, the scale bar length is 50 μm).

**Figure 2 brainsci-14-00445-f002:**
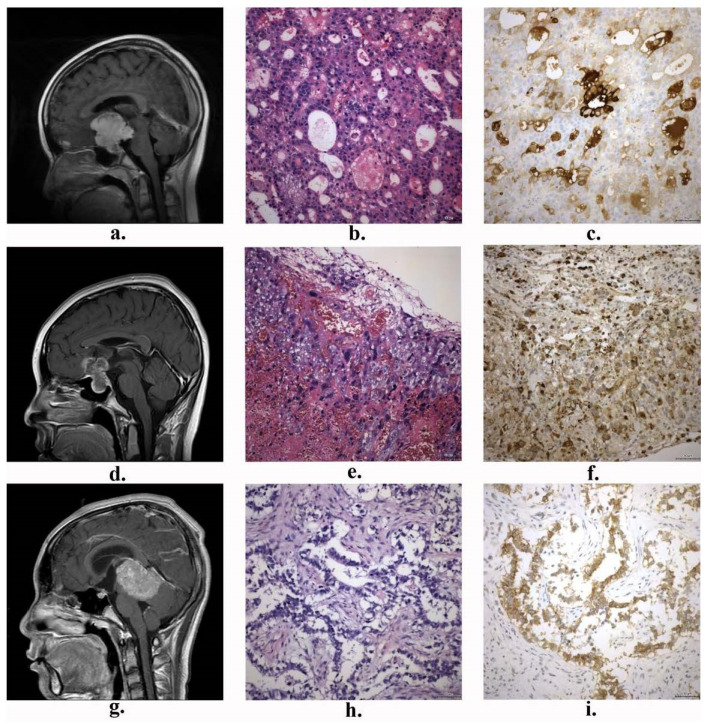
Typical MRI images and histopathological pictures of central nervous system non-germinomatous germ cell tumors: (**a**,**d**,**g**) T1-weighted gadolinium-enhanced MRI from the patients with yolk sac tumor (YST), choriocarcinoma (CC) and embryonal carcinoma (EC); (**b**,**e**,**h**) HE stained images of YST, CC, and EC tissues; (**c**,**f**,**i**) IHC staining images of AFP, HCG, and CD30 expression. (HE and IHC pictures: 200×, the scale bar length is 50 μm).

**Figure 3 brainsci-14-00445-f003:**
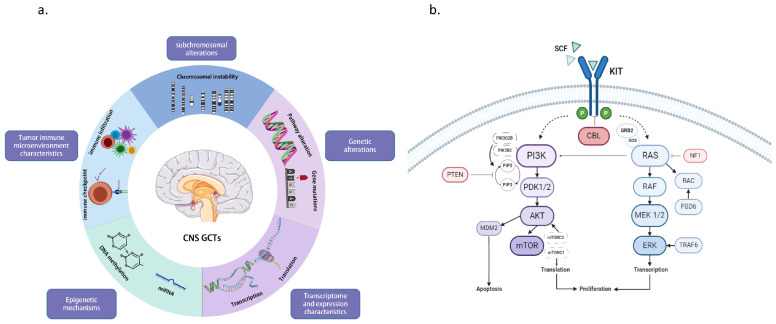
Summary of Molecular Advances in Central Nervous System germ cell tumors: (**a**) Advancements in Guiding Risk Stratification and Future Directions in Central Nervous System Germ Cell Tumors; (**b**) Genes and Signaling Pathways Associated with Central Nervous System Germ Cell Tumors.

**Table 1 brainsci-14-00445-t001:** Summary of chromosomal aberrations in CNS GCTs.

Chromosomal Aberrations	Li B 2024 [32]. *n* = 96 *	Takami H 2019 [13]. *n* = 74 **	Terashima K 2014 [27]. *n* = 27	Fukushima S 2014 [30]. *n* = 65	Wang L 2014 [31]. *n* = 62 ***	Schneider DT 2006 [29]. *n* = 19 ****	Okada Y 2002 [28]. *n* = 25
Gains	1p (9%), 1q (21%), 2p (29%), 2q (28%), 4p (5%), 4q (5%), 6p (7%), 6q (7%), 7p (19%), 7q (18%), 8p (35%), 8q (40%), 12p (38%), 12q (21%), 14q (19%), 15q (8%), 17q (12%), 18p (5%), 20p (21%), 20q (24%), 21p (28%), 21q (47%), 22q (33%)	1q (20%), 2p (12%), 2q (12%), 7p (9%), 7q (9%), 8p (14%), 8q (15%), 12p (15%), 12q (12%), 14q (7%), 15q (5%), 17q (7%), 19p (8%), 19q (8%), 20p (14%), 20q (15%), 21q (28%), 22q (11%), Xp (22%), Xq (22%), Yp (7%)	1q (44%), 2p (37%), 7q (37%), 8q (41%), 12p (59%), 14 (33%), 20q (30%), 21 (63%), 22 (41%), Xq (44%)	1q (46.2%), 12p (44.6%), 21q (66.1%), X (58.5%)	1q, 12p, 14q, 21q, X	1q (47%), 8q (47%), 12p (58%), 21 (32%), X (26%)	12p (20%)
Losses	4p (18%), 4q (18%), 5p (17%), 5q (19%), 6p (7%), 6q (6%), 9p (20%), 9q (21%), 10p (11%), 10q (8%), 11p (16%), 11q (19%), 13q (41%), 15q (6%), 16p (14%), 16q (11%), 17p (12%), 17q (5%), 18p (19%), 18q (19%), 19p (9%), 19q (8%), 20p (5%)	1p (5%), 5p (12%), 5q (15%), 9p (12%), 9q (12%), 11p (9%), 11q (15%), 13q (16%), 15q (5%), 16p (7%), 16q (7%), 18p (8%), 18q (9%), 19q (5%), 22q (7%)	1p (26%), 4q (26%), 5q (33%), 9q (30%), 10q (37%), 11q (41%), 13 (48%)	5q (30.8%), 11q (33.8%), 13q (41.5%)	10q, 11q, 13q, 17p	11q (26%), 13q (11%), 18q (21%)	13q (12%)

*: The incidence of chromosomal aberrations accounting for less than 5% was excluded; **: The incidence of chromosomal aberrations was calculated by dividing the number of cases with the corresponding chromosomal changes by the total number of cases, excluding those with changes accounting for less than 5%; ***: Without specific case numbers; ****: Due to restricted tumor material and lack of normal reference tissue in some cases, no analysis defining the constitutional sex chromosomal status could be performed; Abbreviations: CNS: central nervous system, GCTs: germ cell tumors.

**Table 2 brainsci-14-00445-t002:** Summary of gene mutations and the relevant pathways in CNS GCTs.

Genes Mutations	Fukushima S 2014 [30]. *n* = 65	Wang L 2014 [31]. *n* = 62	Li B 2024 [32]. *n* = 96	Ichimura K 2016 [43]. *n* = 124
KIT	22% (germinomas: 40%, NGGCTs: 6%)	26%	33%	23% (germinomas: 40%, NGGCTs: 11%)
MAPK pathway	RAS: 11% (germinomas: 20%, NGGCTs: 3%)	RAS: 19% (KRAS: 15%; NRAS: 5%)	KRAS: 14%; RRAS2: 7%	RAS: 12% (germinomas: 19%, NGGCTs: 6%)
PI3K pathway	ND	AKT1: 19% *; AKT3: 2% mTOR: 8%	ND	PIK3C2B: 4% (germinomas: 3%, NGGCTs: 2%) PIK3R2: 2% (germinomas: 3%, NGGCTs: 2%) mTOR: 5% (germinomas: 7%, NGGCTs: 6%)
Other genes	ND	CBL: 11% NF1: 3% PTEN: 2%	CBL:5% BCORL1: 7% BRAF: 3% NF1: 3% USP28: 8%	MDM2: 2% (germinomas: 1%, NGGCTs: 0%) PTEN: 2% (germinomas: 3%, NGGCTs: 2%) CBL: 5% (germinomas: 4%, NGGCTs: 7%) NF1: 4% (germinomas: 3%, NGGCTs: 2%) FGD6: 4% (germinomas: 7%, NGGCTs: 2%) TRAF6: 1% (germinomas: 1%, NGGCTs: 2%)

*: AKT 1: Copy number gain. Abbreviations: CNS: central nervous system, GCTs: germ cell tumors, NGGCTs: non-germinomatous germ cell tumors, KIT: transmembrane protein with tyrosine kinase activity, MAPK: mitogen-activated protein kinase, PI3K: phosphoinositide 3-kinase, RAS: Rat Sarcoma (oncogene), KRAS: V-Ki-Ras2 Kirsten Rat Sarcoma 2 Viral Oncogene Homolog; NRAS: Neuroblastoma RAS Viral (V-Ras) Oncogene Homolog, RRAS2: Ras-related protein R-Ras2, AKT: v-akt murine thymoma viral oncogene homolog, AKT1: AKT serine/threonine kinase 1, AKT3: AKT serine/threonine kinase 3, mTOR: Mammalian Target Of Rapamycin, CBL: Casitas B-lineage Lymphoma, NF1: Neurofibromin 1, PTEN: Phosphatase and tensin homolog, BCORL1: BCL6 Corepressor Like 1, USP28: ubiquitin specific peptidase 28, BRAF: B-Raf proto-oncogene, serine/threonine kinase, PIK3C2B: phosphatidylinositol-4-phosphate 3-kinase catalytic subunit type 2 beta, PIK3R2: phosphoinositide-3-kinase regulatory subunit 2, MDM2: Mouse Double Minute 2 Homolog (proto-oncogene), FGD6: FYVE, RhoGEF and PH domain containing 6, TRAF6: TNF receptor associated factor 6.

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
