# Peer review of "CNS Germ Cell Tumors: Molecular Advances, Significance in Risk Stratification and Future Directions"

_brainsci, 2024, doi:10.3390/brainsci14050445_

Round 1

Reviewer 1 Report

Comments and Suggestions for Authors

The authors present an extensive review of the current knowledge regarding molecular and genetic alterations within the whole spectrum of central nervous system germ cell tumors.  This is an excellently written review article. 

 I have 1 major concern which can be easily addressed: The title of the article does not reflect the reality of the manuscript.  Although they do a spectacular job with the molecular advances, there is no true correlation between what has been seen in the laboratory, risk stratification, or clinical practice.  At best the authors point in multiple directions for possible future clinical investigation.  A clinician looking at this title will be disappointed in not finding any pragmatic advice for current management.  I would suggest perhaps changing the title to "CNS germ cell tumors: Molecular advances and future directions"

Reviewer 2 Report

Comments and Suggestions for Authors

Germ cell Tumors of the CNS are rare tumors. This is well-written summary of molecular changes detected in these tumors.

- The addition of an MRI Scan and histopathology pictures would improve the manuscript.

- The authors should provide in some parts of the described data (i.e. chromosomal instatbility) the percentage of cases presenting a specific change

- The authors should describe in more detail which mutations were found in which germ cell tumor type, i.e. germinoma, yolk sac tumor etc.

Comments on the Quality of English Language

minor improvements

Reviewer 3 Report

Comments and Suggestions for Authors

The manuscript entitled "CNS Germ Cell Tumors: Molecular Advances and Significance in Risk Stratification and Clinical Practice" by Jiajun Zhou provides a comprehensive review of CNS GCTs with an emphasis on molecular genetics. However, it requires major revisions to enhance clarity, depth, and scholarly rigor.

Major comments:

1.          The introduction lacks a clear, concise summary of existing knowledge and the value added by this work. It is recommended to restructure this section to outline the clinical importance of CNS GCTs, succinctly summarize the existing diagnostic and treatment challenges, and clearly state the review’s objectives.

2.          A detailed discussion on the limitations of current treatments would benefit the review. These should be presented in bullet points for clarity, followed by a comprehensive discussion incorporating recent studies and clinical trials for support.

3.          The paper lacks a thorough overview of recent clinical trials, especially those that failed, and their implications for survival benefits. Including this analysis will align with the paper’s themes and provide a critical evaluation of past approaches, informing future research directions.

4.          The manuscript should incorporate figures, tables, or flowcharts to better illustrate key points, such as the pathways involved in tumor pathogenesis or summaries of the molecular alterations discussed. Visual aids are crucial for helping readers easily grasp complex information, which currently appears dense and technical.

5.          Some sections require significant technical corrections to enhance accuracy and incorporate the latest information. For example, the advancements in targeted therapy and their clinical trial outcomes should be updated to reflect the most recent findings.

6.          Several statements lack up-to-date references or citations entirely. It is crucial to ensure that all significant claims are supported by recent literature, enhancing the manuscript's credibility and scholarly value.

Minor Comments:

7.          The manuscript would benefit from proofreading to correct grammatical errors and improve language fluency.

8.          Ensure consistent use of terminology when describing molecular pathways and genetic alterations.

9.          The authors could further explore the implications of genetic insights on personalized medicine approaches, particularly how these insights could influence future clinical practices.

Comments on the Quality of English Language

The manuscript would benefit from proofreading to correct grammatical errors and improve language fluency.

Round 2

Reviewer 3 Report

Comments and Suggestions for Authors

I appreciate the effort and detail you put into addressing all the concerns raised. The updates and clarifications have significantly strengthened your paper, and I believe it will be a valuable addition to the journal.